# Sustainable One-Step Solid-State Synthesis of Antibacterially Active Silver Nanoparticles Using Mechanochemistry

**DOI:** 10.3390/nano10112119

**Published:** 2020-10-25

**Authors:** Mária Kováčová, Nina Daneu, Ľudmila Tkáčiková, Radovan Búreš, Erika Dutková, Martin Stahorský, Zdenka Lukáčová Bujňáková, Matej Baláž

**Affiliations:** 1Department of Mechanochemistry, Institute of Geotechnics, Slovak Academy of Sciences, Watsonova 45, 04001 Košice, Slovakia; dutkova@saske.sk (E.D.); stahorsky@saske.sk (M.S.); bujnakova@saske.sk (Z.L.B.); 2Advanced Materials Department, Jozef Štefan Institute, Jamova 39, 1000 Ljubljana, Slovenia; nina.daneu@ijs.si; 3Department of Microbiology and Immunology, University of Veterinary Medicine and Pharmacy, Komenského 73, 04181 Košice, Slovakia; ludmila.tkacikova@uvlf.sk; 4Institute of Materials Research, Slovak Academy of Sciences, 04001 Košice, Slovakia; rbures@saske.sk

**Keywords:** mechanochemistry, green synthesis, silver nanoparticles, antibacterial activity, plants, solid-state synthesis

## Abstract

A combination of solid-state mechanochemical and green approaches for the synthesis of silver nanoparticles (AgNPs) is explored in this study. *Thymus serpyllum* L. (SER), *Sambucus nigra* L. (SAM) and *Thymus vulgaris* L. (TYM) plants were successfully applied to reduce AgNO_3_ to AgNPs, as confirmed by X-ray diffraction analysis, with SER being the best reducing agent, and TYM being the worst. The experiments were performed via a one-step planetary milling process, where various AgNO_3_:plant mass ratios (1:1, 1:10, 1:50 and 1:100) were investigated. Atomic absorption spectrometry indicated that the stability of the mechanochemically produced AgNPs increased markedly when a sufficiently large quantity of the reducing plant was used. Furthermore, when larger quantities of plant material were employed, the crystallite size of the AgNPs decreased. TEM analysis revealed that all AgNPs produced from both AgNO_3_:plant ratios 1:1 and 1:10 exhibit the bimodal size distribution with the larger fraction with size in tens of nm and the smaller one below 10 nm in size. The antibacterial activity of the produced AgNPs was observed only for AgNO_3_:plant ratio 1:1, with the AgNPs prepared using SER showing the greatest antibacterial properties.

## 1. Introduction

Silver nanoparticles (AgNPs) are studied extensively for their application in electronics, photonics, photocatalysis, food technology, textile and pharmacological industries, environmental safety and many other fields [1,2,3,4,5]. They are also used in health care and biomedicine because of their antibacterial [6], antiviral [7], antifungal [8], anti-inflammatory [9], antiangiogenic [9] and anticancer [10] properties.

There are three key methodologies employed for the production of AgNPs: physical, chemical and biological [2,3,4]. The last one is of particular interest as harmful synthetic reducing agents are replaced by non-toxic molecular species (proteins, carbohydrates, antioxidants, starch, etc.) which are produced by bacteria, yeasts, fungi as well as plants [10,11,12]. A myriad of plants have been investigated for the green (biological) synthesis of AgNPs [2,13], including *Thymus vulgaris* L. (TYM) [14,15], *Thymus serpyllum* L. (SER) [16] and *Sambucus nigra* L. (SAM) [17,18]—all of which are common, easily accessible plants. However, AgNPs produced in this way are not very stable, and higher temperatures are usually required for the preparation of the plant extract, as well as for the synthetic reduction step [19,20,21,22].

In this article, we propose a one-step solid-state biomechanochemical synthesis of AgNPs using AgNO_3_ as the Ag^0^ precursor and three common plants as reducing agents. The efficiency of the synthesis process depends on the plant employed and the AgNO_3_:plant mass ratio used in the reaction. AgNPs produced via this biomechanochemical route, using the AgNO_3_:plant mass ratio 1:1 exhibit antibacterial activity. To the best of our knowledge, such a comparative study on a biomechanochemical synthesis using different plants has not been reported to date. Moreover, only few reports are available discussing AgNPs synthesis by ball milling a metal salt with natural materials [23,24,25,26,27,28,29,30].

## 2. Materials and Methods

### 2.1. Materials

AgNO_3_ (99.9% purity from Centralchem, Banská Bystrica, Slovakia) was used without any further purification. *Thymus vulgaris* L., *Sambucus nigra* L. and *Thymus serpyllum* L. plants were collected in summer from the meadow on the University of Veterinary Medicine and Pharmacy campus in Košice, Slovakia, and dried up to a constant weight in the dark at room temperature.

### 2.2. Mechanochemical Synthesis

During the mechanochemical synthesis, a total solid mass of 3.0 g was milled in a Pulverisette 7 Premium line planetary ball mill (Fritsch, Idar-Oberstein, Germany) under the following conditions: air atmosphere, 15 tungsten carbide (WC) milling balls (10 mm diameter), ball-to-powder ratio 37, milling speed 500 rpm and milling time 2.0 h. Four different AgNO_3_:plant mass ratios (1:1, 1:10, 1:50 and 1:100) were tested. Hereafter, the AgNO_3_:plant mixtures are denoted as Ag:plant (Ag:TYM/SAM/SER). A 1.0 g sample of the as-obtained powder (labeled as Ag:plant M) was washed with 100 mL distilled water to remove non-reacted AgNO_3_ and non-stabilized Ag^0^ (these samples are labeled as Ag:plant W).

### 2.3. Characterization

X-ray diffraction (XRD) patterns of the samples were obtained using a D8 Advance diffractometer (Bruker, Billerica, MA, USA) with CuKα radiation of 0.15407 nm wavelength at 40 kV accelerating voltage and 40 mA electric current. The samples were scanned over a 15°–70° range of diffraction angle (2θ). Rietveld refinement of the XRD data (using a TOPAS Academic software by Coelho Software, Brisbane, Australia) was used to determine the phase composition, crystallite size and unit cell parameters of AgNPs in the samples after washing.

The level of stabilization of the mechanochemically prepared AgNPs (obtained as solid samples after washing) was analyzed using an atomic absorption spectrometer SPECTRAA L40/FS (Varian, Crawley, UK) by measuring the content of silver. Prior to the measurement, the samples were dried at 50–60 °C for 1 h, and subsequently, 0.1 g was dissolved in 10 mL concentrated nitric acid. The solution was then diluted with distilled water to obtain the volume of 100 mL.

The grain size analysis was performed using a particle size laser diffraction analyzer Mastersizer 2000E (Malvern, Malvern, UK) in the dry mode. Each sample was measured three times.

UV-Vis spectra were collected using the UV-Vis spectrophotometer Helios Gamma (Thermo Electron Corporation, Waltham, Massachusetts, USA) working in the range 200–800 nm in a quartz cell by dispersing synthesized particles in absolute ethanol by ultrasonic stirring for 10 min.

Transmission electron microscopy (TEM) analyses of the samples with AgNO_3_:plant ratio of 1:1 and 1:10 were performed using a 200-kV microscope JEM 2100 (JEOL, Akishima, Japan) with LaB_6_ electron source and equipped with an energy dispersive X-ray spectrometer for chemical analyses. For TEM analyses, the powdered samples were ultrasonically dispersed in ethanol for a few minutes; then, a droplet of the suspension was applied onto a lacey carbon-coated copper grid and dried. The grids were additionally coated with a thin layer of carbon for improved stability of the samples under the electron beam.

Particle size distribution of the filtrate was measured by a photon cross-correlation spectroscopy using a Nanophox particle size analyzer (Sympatec, Clausthal-Zellerfeld, Germany). A portion of each nanosuspension was diluted with distilled water to achieve a suitable concentration for the measurement. This analysis was performed using a dispersant with the refractive index of 1.33. The measurements were repeated 3 times.

### 2.4. Antibacterial Activity

The antibacterial properties of the samples were evaluated by the agar well diffusion method. The tested bacteria (*Staphylococcus aureus* CCM 4223 and *Escherichia coli* CCM 3988) were obtained from the Czech collection of microorganisms (CCM). A few hours before the antibacterial activity tests, the powders containing AgNPs were mixed with the distilled water (20 mg of the powder was put into 1 mL water) and sonicated. Just before the inoculation on the agar plate, the suspensions were mixed again using a laboratory stirrer to ensure the most homogeneous distribution of the solid particles in water. Subsequently, 50 µL of samples prepared in the form of a suspension were introduced into the wells and gentamicin sulphate (Sigma-Aldrich, Saint-Louis, Missouri, USA), with a concentration of 30 mM, was used as a positive control. The antibacterial activity is expressed as the relative inhibition zone diameter (RIZD), where the activity of the positive control is considered as 100% and the activity of the tested substances compared with it. Other details of antibacterial activity evaluation in this study are the same as reported previously [26].

## 3. Results and Discussion

### 3.1. X-ray Diffraction Analysis

#### 3.1.1. As-Received Powders

Powder X-ray diffraction (XRD) was used to observe the formation of AgNPs for various AgNO_3_:plant mass ratios. The XRD patterns for the as-obtained products, together with the XRD patterns of the pure plants are compared in Figure 1a–c. In the XRD pattern for the Ag:TYM M 1:1 sample (Figure 1a), diffraction peaks representing both AgNO_3_ and Ag^0^ can be seen. The Ag^0^ diffraction peaks can be well-indexed to the face-centered cubic structure of Ag. It can be noted from the XRD pattern that the AgNO_3_ phase is dominant; thus, it can be suggested that only partial Ag^+^ → Ag^0^ reduction after 2.0 h of milling occurred. Upon increasing the plant amount to Ag:TYM M 1:10, the AgNO_3_ peaks are no longer visible whilst the peaks associated with Ag^0^ are the most intensive. This sample also contains WC peaks assumed to be obtained from wear of the milling media used. They can be also found in the patterns of Ag:TYM M 1:50 and 1:100 samples and their intensity seems to decrease with increasing plant amount. It must be noted that the Ag:TYM M 1:1 sample is an exception to this, which might be associated with the fact that the reaction does not reach completion. A small amount of AgCl, indicated by the presence of a diffraction peak at 32°, was also detected in the Ag:TYM M 1:10 sample. Its formation can be explained by the reaction of Ag^+^ ions with Cl^−^ ions present in the plant matrix. The peaks of AgCl are detectable for both Ag:TYM M 1:50 and 1:100 samples. In these samples, the TYM mass is too high to identify any characteristic Ag^0^ peaks; in fact, even the most intensive peak at ca. 38° cannot be clearly identified in the observed amorphous halo of the plant matrix. All of the Ag:TYM M samples were found to have an additional diffraction peak at ca. 26.6°, whose intensity is enhanced with ascending TYM content. Together with other weaker diffractions at 20.9° and 50.0°, the observed phase could be indexed to a hexagonal modification of quartz SiO_2_ (ICDD no. 77-1060), which is present in TYM plant.

In Figure 1b, the XRD patterns of Ag:SAM M samples are displayed. A small amount of AgNO_3_ appears to still be present in the sample Ag:SAM M 1:1; however, its peak intensities are weak, whereas the peaks representing Ag^0^ are dominant. This is different from Ag:TYM M 1:1 (Figure 1a), where very intense AgNO_3_ peaks are exhibited. Similarly to the Ag:TYM M 1:10 sample, WC peaks are also present in the Ag:SAM M 1:10 sample, but at much higher intensity, as all three main peaks located at 31.4°, 35.5° and 48.1° are clearly seen. A small amount of wear was also detected for the SAM-rich M samples (Ag:SAM 1:50 and Ag:SAM 1:100). It seems that the amount of wear increases with the amount of AgNO_3_ introduced into the Ag:SAM M system, similarly to Ag:TYM M. Again, the Ag:SAM M 1:1 sample does not follow this trend. The peaks representing AgCl and SiO_2_ were not observed for Ag:SAM M samples.

In the XRD pattern of Ag:SER M 1:1 sample (Figure 1c), the peaks of AgNO_3_ are barely detectable. Diffraction peaks of Ag^0^ are the most intensive in this XRD pattern. The peaks of SiO_2_ (the most intensive one at 26.6°) are of lower intensity upon comparison with Ag:TYM M samples. This is a result of the fact that the content of SiO_2_ in the SER plant seems to be lower than in TYM. As with all other Ag:plant M 1:10 samples, no diffraction peaks representing AgNO_3_ are visible for the Ag:SER M 1:10 sample. Unlike the other Ag:plant M 1:10 samples, WC impurity was not detected in an Ag:SER M 1:10 sample. However, relatively intense reflections corresponding to WC can be seen in Ag:SER M 1:50 and 1:100 samples, with the former exhibiting the greatest intensity. Similarly to Ag:TYM M samples, AgCl was evidenced in small amounts in the Ag:SER M 1:10 sample and in a significant amount in the two SER-richest M samples.

We observed differences in the reducing ability of the three used plants. The green synthesis of AgNPs is based on the reduction in metal ion (Ag^+^) to its elemental form (Ag^0^). Plants contain a large number of active metabolites with reducing ability from the groups of terpenoids, flavonoids, tannins, phenolic acids, saponins, steroids, alkaloids, saccharides, proteins, amino acids, enzymes or vitamins. The mentioned substances contain reducing groups (OH, CH=O, NH_2_, SH) which are all able to reduce Ag^+^ ions and produce the AgNPs, but also play an important role in their stabilization [12,31]. We suppose that a different performance of each plant is connected with a various content of reducing biomolecules. The differences in the biomechanochemical AgNPs synthesis rate have also been recently observed when using different lichens [29,32].

To shed more light on the formation of the AgCl phenomenon, the informative region (from 20° to 50°) from the XRD patterns of the two TYM- and SER-richest M samples is depicted in Figure 1d. In all these samples, a significant amount of AgCl was observed, being more pronounced for Ag:SER M samples. Its content seems to be more-or-less similar for Ag:SER M 1:50 and 1:100 samples; however, the peak corresponding to Ag^0^ was not detected for the latter SER-richest M sample, whereas a small one could be detected for the Ag:SER M 1:50 sample. It means that AgCl is formed preferentially—i.e., the supplied Ag^+^ ions react with available Cl^−^ ions at first, and after they are used up, the reduction process leading to AgNPs formation begins. In the case of Ag:SER M 1:100, the supplied amount of Ag^+^ ions was just enough for AgCl formation but was not satisfactory for the reduction process to start. *Thymus serpyllum* L. appears to contain larger amounts of Cl^−^ than *Thymus vulgaris* L., which results in the higher amount of AgCl forming, but a lower amount of Ag^0^ in this case. The larger quantity of AgNPs formed when using TYM is clear from the comparison of the intensities of Ag^0^ peak both for 1:50 (clearly higher intensity for Ag:TYM M than for Ag:SER M sample) and 1:100 samples (no peak for Ag:SER M, whereas at least small one for Ag:TYM M). When the supplied Ag^+^ ion content is sufficient (e.g., as with the Ag:plant M 1:1 samples), the overall amount of the reducing agents in the plant, being significantly higher for SER than TYM, becomes most important and significant Ag^+^ → Ag^0^ reduction takes place.

To justify using plants as reducing agents, milling of pure AgNO_3_ under the same experimental conditions was also performed; however, we did not observe any chemical changes in this case (see Appendix A: XRD pattern of milled AgNO_3_ in the Electronic Appendix A).

#### 3.1.2. Washed Powders Subjected to Rietveld Refinement

After thorough washing with distilled water, the XRD patterns of Ag:plant 1:1 and 1:10 samples were measured again and subjected to Rietveld refinement, with the main aim to calculate the potential content of impurities and the crystallite size. The obtained XRD patterns are shown in Figure 2, and the Rietveld refinement results are summarized in Table 1, Table 2 and Table 3. The three most intensive peaks in the Ag:plant 1:1 samples were indexed with (111), (200) and (220) crystallographic planes of face-centered cubic Ag^0^. These diffractions were found in all XRD patterns of the samples prior to washing (Figure 1) where the reduction process was at least partly successful.

The phase analysis results (Table 1) show that the produced AgNPs are of high purity for all Ag:plant 1:1 samples. In addition to Ag^0^, only a very small amount of WC contamination is evidenced for the Ag:TYM sample as further confirmed by the XRD patterns in Figure 2. In the case of washed Ag:plant 1:10 samples, the peaks of other crystalline phases not corresponding to Ag^0^ can be clearly seen. The greatest purity of AgNPs (of 93.6%) was evidenced for the Ag:SER 1:10 sample, where 6.4% of AgCl was formed as an admixture and no WC wear was evidenced. The WC contamination is higher for Ag:SAM and Ag:TYM 1:10 samples, being as high as 62.3% for the former. In the case of Ag:TYM 1:10, in addition to Ag^0^ and WC, AgCl is also present in a small amount (3.4%).

The crystallite size of all Ag:plant 1:1 samples is larger than that of 1:10 samples (Table 2). The higher amount of reductive agents in the latter case leads to the formation of smaller crystallites, which most probably results in more effective stabilization [33,34].

For the Ag:TYM system, the difference between 1:1 and 1:10 samples seems to be the least significant (the calculated crystallite size for Ag:TYM 1:1 and 1:10 samples was 22 and 19 nm, respectively), upon comparison with the other systems, most probably due to a limited reduction progress. Similar crystallite size values were reported in the other study [14] on AgNPs synthesis using the same plant.

A significantly larger difference in crystallite size was found for Ag:SAM 1:1 system (the calculated crystallite size for Ag:SAM 1:1 and 1:10 samples was 18 and 13 nm, respectively), which is in agreement with other reports using similar plants [17,18].

From the three Ag:plant 1:1 samples, the largest crystallites of 27 nm in size were obtained for the Ag:SER 1:1 sample. Thus, the crystallite size for Ag:plant 1:1 samples seems to correlate with the reaction progress (the largest amount of available reducing species leads to the most pronounced growth of AgNPs). For the Ag:SER 1:1 sample, a microstrain of 0.2717 ± 0.0830 was also found to contribute to the peak broadening of Bragg’s peaks of Ag^0^. For the Ag:SER 1:10 sample, the reduction of crystallite size to 19 nm was observed after using 10 times more plant material.

In general, using a larger number of plants results in a more effective reduction and formation of smaller Ag^0^ crystallites [33,34,35,36,37]. Furthermore, under these conditions, Ag^+^ ions are divided efficiently into small groups and the reaction propagates at more sites simultaneously. Similar results were reported in the article of Shaik et al. [38], where increasing the dosage of *Origanum vulgare* L. extract was shown to directly control the crystallite size of AgNPs. Thus, it can be claimed that the AgNO_3_:plant mass ratio plays a crucial role in determining the reaction progress.

Finally, the volume of the face-centered cubic unit cell of Ag was calculated for all Ag:plant 1:1 and 1:10 samples (Table 3). The crystal lattice appears to be more compact for the Ag:plant 1:1 samples than in the case of Ag:plant 1:10 ones. The unit cell volume decreases in the following order Ag:TYM > Ag:SAM > Ag:SER, which is reflective of the reaction progress observed. For Ag:plant 1:10 samples, the reduction in the cell is not very pronounced; however, the greatest decrease in unit cell volume is observed for the Ag:SER sample.

### 3.2. Atomic Absorption Spectrometry Analysis

The silver content in the solid samples (both as-obtained after milling (M) and after washing (W) with distilled water) was analyzed by atomic absorption spectrometry (Table 4). The W/M ratio represents the stability of the prepared AgNPs—i.e., the higher the value, the better the stability of the AgNPs in the plant matrix. In the Ag:plant 1:1 samples, the W/M ratio is <1. Upon increasing the plant dose, W increases which, leads to an increase in the W/M ratio (>1, this is the case for all Ag:plant 1:10, 1:50 and 1:100 samples).

When comparing the same AgNO_3_:plant mass ratios, but varying the plant, W/M is consistently higher for the Ag:SER samples (with the exception of Ag:plant ratio 1:100), which indicates the formation of the most effectively stabilized AgNPs in this case. With this in mind, two different scenarios are possible (Figure 3):In the Ag:plant 1:1 samples, the concentration and subsequent dissolution (into distilled water) of non-stabilized Ag^0^ and Ag^+^ is relatively high (Figure 3a). The water-soluble plant substances are washed away with the unstabilized Ag species (meaning not retained in the powder, but they are stabilized by the water-soluble components of the plant) as well, but their dissolution from the Ag:plant 1:1 samples is not as high as in the other cases.In plant-rich samples, only stabilized Ag^0^ is present; thus, only water-soluble plant species are washed away, resulting in a larger amount of Ag being identified in the samples (Figure 3b).

### 3.3. Grain Size Analysis

The washed samples in all four tested ratios were subjected to grain size analysis in the range of micrometers (Figure 4). In all cases, small nanoparticles with crystallite size in tens of nanometers (Table 1) are agglomerated into larger micron-sized grains. In all samples, a trend of gradual increase in the grain size with the amount of plant introduced was observed until the ratio 1:50 (except the sample Ag:SER 1:50). Afterwards, the grain size decreased for Ag:plant 1:100 samples. The largest difference in grain sizes between largest and smallest Ag:plant ratio was registered for Ag:SAM samples. Namely, the biggest and the smallest grains with size 233 and 65 µm were evidenced for Ag:SAM 1:50 and 1:1 ratios, respectively.

The observed grain size is not to be confused with the crystallite size detected by X-ray diffraction (also described as particle size, namely when discussing TEM results). In mechanochemical synthesis, we usually obtain nanocrystalline materials; however, the individual nanoparticles are agglomerated to micron-sized grains. Whereas TEM, UV-Vis and XRD report the results of crystallite size of individual nanoparticles, SEM, PCCS or DLS methods report the size of grains/clusters/agglomerates (Figure 5). These two approaches are often incorrectly interchanged in the papers.

### 3.4. UV-Vis Results

In Figure 6, the UV-Vis spectra of the washed Ag:plant 1:1 samples are presented. In the spectrum of Ag:TYM 1:1, two key absorbances can be identified: one at 352 nm, which corresponds to the plant matrix and another at 454 nm, which is characteristic for AgNPs, thus supporting the conclusion that the synthesis was successful. Similarly, Heidari et al. [15] reported an absorbance at 440 nm when employing the same plant, TYM. In the case of the two other plants, only the surface plasmon resonance peak was identified, characteristic of AgNPs, with absorbance maxima obtained at 440 nm for Ag:SER and 436 nm for Ag:SAM. Upon comparison with the results reported by Erci and Torlak [16], where the authors report an absorption peak for AgNPs (prepared using SER) at 467 nm, herein, the absorption band is observed at slightly lower wavelengths. The absorbance maximum for the Ag:SAM system is slightly higher in wavelength than the one reported by Moldovan et al. [18] (407 nm). The phenomenon that smaller nanoparticles should possess the SPR band at the lower wavelength was at least partially confirmed for the Ag:SAM 1:1 sample, which contained the smallest NPs (18 ± 0.3 nm), and the SPR band is located at 436 nm (the lowest wavelength of the studied samples). However, this correlation did not work for the other two Ag:plant 1:1 systems, as the largest particles evidenced in the case of the Ag:SER 1:1 sample (27 ± 3 nm) exhibited the SPR band located at 440 nm, and in the case of Ag:TYM 1:1, where the crystallite size was 22 nm (in between those of Ag:SER 1:1 and Ag:SAM 1:1), the maximum SPR value from all three studied samples located at the highest wavelength (454 nm) was observed. The assumption about increasing the wavelength of SPR maximum in the UV-Vis spectra was also not confirmed when comparing the results from relevant literature; as for AgNPs prepared using TYM, SAM and SER, the SPR peak position was 440, 407 and 467 nm, and the crystallite size calculated from TEM was 30, 25 and 25.2 nm, respectively [15,16,18]. However, the validity of such a comparison of the results from different studies achieved at different experimental setups is questionable.

### 3.5. TEM Analysis

The size and morphology of AgNPs prepared from AgNO_3_:plant ratios 1:1 and 1:10 after washing were analyzed by TEM (Figure 7). Selected area electron diffraction (SAED) was used to confirm the presence of AgNPs in all samples (see Appendix A: SAED patterns of the samples with Ag:plant ratio (a) 1:10 and (b) 1:1 in the Electronic Appendix A), where the main crystallographic planes of Ag^0^ are also indexed. All SAED patterns are ring-patterns typical for diffraction from randomly oriented nanoparticles and contain only reflections belonging to the face-centered Ag^0^ phase. TEM images show that all AgNPs are embedded in the organic matrix. It is evident that the average size of the AgNPs is smaller in the samples prepared from a AgNO_3_:plant ratio 1:10 (Figure 7a), whereas the presence of larger AgNPs is observed in all samples prepared from a ratio 1:1 (Figure 7b), which is in accordance with the XRD analyses (Figure 2). The average particle size is always larger for Ag:plant 1:1 samples than for 1:10 samples, and no difference was observed when using different plants. Namely, the particle sizes range from <5 nm to around 40 nm in samples with 1:10 ratio, whereas in the samples with 1:1 AgNO_3_:plant ratios, the largest AgNPs reach diameters around 80 nm. This type of particle size distribution, including very small NPs and larger fraction reaching several tens of nm, is typically observed in AgNPs synthesized using a mechanochemical approach [26,29]. The presence of very small Ag^0^ crystallites with sizes <5–10 nm cannot be detected by XRD and supports the use of a combination of techniques for characterization of samples prepared by this synthesis approach. It indicates that the reduction of AgNO_3_ is triggered upon contact with the plant matrix containing the necessary reducing agents. The growth of larger AgNPs occurs by Ostwald ripening, where the smaller particles tend to grow into larger, energetically more stable particles. The particle size is limited by the initial concentration of AgNO_3_ and the amount of the reducing agents. The synthesized AgNPs are stabilized by the organic matrix. It is interesting to note that the fraction of the smallest AgNPs is significantly lower in the Ag:SER 1:10 sample (see the inset of the sample Ag:SER 1:10 in Figure 7a), which correlates with the most pronounced reactivity of all Ag:plant systems. The presence of parallel twins was observed in some of the larger AgNPs in the 1:1 samples. This type of twinning is typical for green-synthesized AgNPs [25] and could be related to the presence of chlorine [39].

The chemical composition of the samples with AgNO_3_:plant ratio 1:10 was analyzed by energy dispersive X-ray spectroscopy (EDS), and the results are shown in Figure 8. The aliquot of sample to be tested consisted of the organic matrix and the AgNPs. A common feature of all the spectra presented herein, is the intense signal arising from Ag, whereas the spectral differences arise from elements most likely stemming from the organic matrix. The intensity of these signals slightly differs for each analyzed area. Typically observed for the Ag:TYM and Ag:SER prepared AgNPs is the presence of considerably intense Si signal, whereas a signal representing P is characteristic of the Ag:SAM sample. The Si signal may correlate with the presence of quartz, as was detected in the XRD patterns for those samples prepared under high concentrations of the TYM and SAM plants (Figure 1). Other detected elements include Mg, Al, S and Cl.

### 3.6. Characterization of the Filtrate of Ag:SER 1:1 Sample after Washing

In order to investigate whether also the non-stabilized elemental silver is being washed out into water together with the non-reacted silver nitrate, the filtrate of the Ag:SER 1:1 sample (selected because of the most advanced reaction progress) was analyzed in terms of photon cross-correlation and UV-Vis spectroscopy (Figure 9a,b, respectively).

The grain size distribution (Figure 9a) showed a small number of grains present in the filtrate; however, the majority of them exhibited a size above 1 micrometer (the mean grain size was equal to 6778 nm). Nevertheless, a small number of nanoparticles arranged into grains of size range 80–160 nm were detected, but their content does not seem to be significant. The UV-Vis spectrum (Figure 9b) did not show the presence of surface plasmon resonance (SPR) suggesting the absence of AgNPs (on the contrary to Figure 6 where the UV-Vis spectra of the washed powders are presented).

A similar result has been obtained from the XRD measurement of the dried filtrate, as the diffraction peaks of elemental silver are absent (Figure 10). Only those ascribed to silver nitrate have been identified, which points to the fact that although the reaction seemed to be complete according to X-ray diffraction of the reaction mixture (Figure 1), there was still a small amount of unreacted silver nitrate present. It also seems that the actual content of the non-stabilized elemental silver being washed out (outlined in Figure 3) is low.

### 3.7. Antibacterial Activity

The antibacterial activity of AgNPs was tested on two bacterial strains—one Gram-negative (*Escherichia coli*) and one Gram-positive (*Staphylococcus aureus*) strain (Figure 11). For comparison, and as a control experiment, the antibacterial effect of AgNO_3_ was investigated, employing the same amount of AgNO_3_ as was introduced into the milling chamber. From the washed products, only Ag:plant 1:1 samples were found to be antimicrobially active. This is because the number of formed AgNPs decreased with the number of introduced AgNO_3_. Thus, in the samples where the amount of plant material is 10-, 50- and 100-fold higher (in the Ag:plant 1:10, 1:50 and 1:100 samples), the content of AgNPs is too low to show antibacterial properties (for reference, please check the content of silver in these samples detected by AAS provided in Table 4). Figure 11 highlights that sample Ag:SER 1:1 is most effective antibacterial agent. The activity of the Ag:SAM 1:1 sample was lower, whilst the Ag:TYM 1:1 sample was the least effective. These findings are in accordance with the reaction progress observations discussed earlier (being the best for Ag:SER system and the worst for Ag:TYM system). It is known that the antibacterial activity of AgNPs is also governed by their size. The formation of smaller AgNPs usually leads to better activity [40,41,42]. However, in the present case, Ag:plant 1:10 samples showed lower activity than Ag:plant 1:1 ones, despite significantly smaller crystallite size (Table 2), but this is most probably related to the low content of AgNPs in Ag:plant 1:10 samples. It might seem surprising that when comparing the antibacterial activity of three Ag:plant 1:1 samples, the one with the largest NPs (Ag:SER) is the best agent, but again, the amount of Ag in this washed sample is the highest here (Table 4), so it seems that not the size but the content of Ag is decisive in our case.

Furthermore, we observe better antibacterial activity against Gram-positive *S. aureus*, which is in accordance with the results of Erci and Torlak [16], but contrary to those reported by Jafari et al. [14]. There are also other studies that report the better activity of AgNPs against Gram-negative bacteria [43,44,45,46]. Interestingly, the antibacterial activity for AgNO_3_ solutions does not decrease proportionally with decreasing the salt concentration. AgNO_3_ outperformed the synthesized Ag:plant samples in all the cases; however, it was administered in the form of solution, whereas our products were introduced as nanosuspensions.

## 4. Conclusions

A simple one-step solid-state approach for preparing stable AgNPs via the mechanochemical reaction of AgNO_3_ with the reducing agents from three well-accessible plants as reducing agents is described in this study. During the milling of AgNO_3_ and the plant, Ag^+^ → Ag^0^ reduction proceeds successfully and the biogenic species released from plants promptly stabilize AgNPs. Increasing the dose of the plants leads to a more rapid reduction reaction and formation of smaller AgNPs. This process was found to be most effective in the case of *Thymus serpyllum* L, and the size of the nanoparticles was also largest when using this plant. Washing the powders after milling with distilled water leads to the effective removal of non-reacted AgNO_3,_ non-stabilized Ag^0^ and water-soluble plant species. The successful formation of AgNPs is corroborated by the surface plasmon resonance peak in the UV-Vis spectra and by TEM analysis clearly showing that the AgNPs were successfully embedded in the plant matrices. The bimodal size distribution with the larger fraction of nanoparticles in tens of nanometers and the smaller one below 10 nm in size was found. The amount of smaller particles was much larger for Ag:plant 1:10 samples. The products prepared using AgNO_3_:plant 1:1 mass ratio exhibited antibacterial activities against both Gram-negative and Gram-positive bacteria. AgNPs produced using *Thymus serpyllum* L. showed the best antibacterial properties. The proposed approach provides an opportunity to overcome the problems associated with the instability of AgNPs prepared in the form of a nanosuspension by a classical green synthetic method. The present study emphasizes a sustainable and environmentally friendly manner of conducting solid-state mechanochemical synthesis.

## Figures and Tables

**Figure 1 nanomaterials-10-02119-f001:**
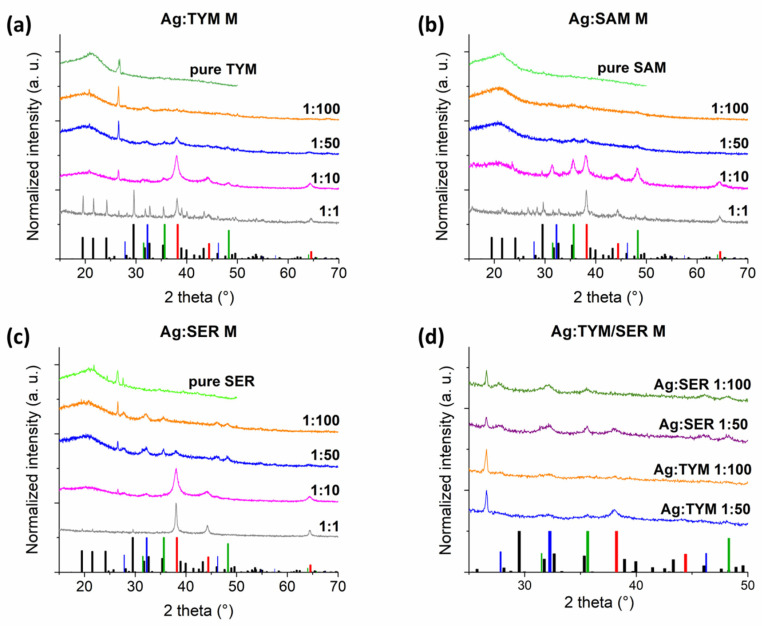
XRD patterns of AgNPs embedded in the plant matrix and that of the pure plants after milling; (**a**) Ag:TYM M; (**b**) Ag:SAM M; (**c**) Ag:SER M; (**d**) the zoomed region between 27° and 50° for the TYM- and SER-rich M samples (bars at the bottom refer to XRD of ICDD PDF2 database: AgNO_3_ (black, 074-2076), Ag (red, 65-2871), WC (green, 51-0939) and AgCl (blue, 71-5209)).

**Figure 2 nanomaterials-10-02119-f002:**
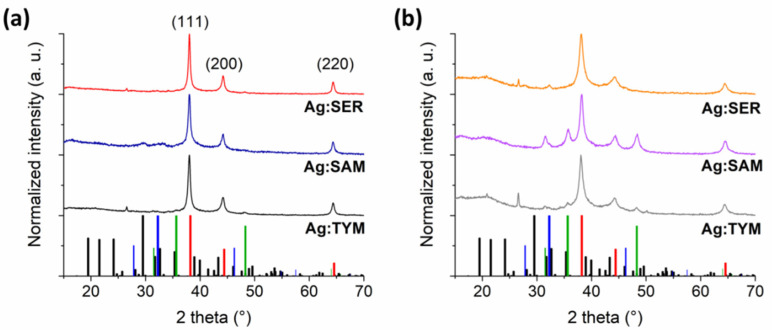
XRD patterns of washed Ag:plant samples; (**a**) 1:1 (three main diffractions of the formed face-centered cubic Ag^0^ are indexed), and (**b**) 1:10 (bars at the bottom refer to XRD of ICDD database: AgNO_3_ (black, 074-2076), Ag (red, 65-2871), WC (green, 51-0939) and AgCl (blue, 71-5209)).

**Figure 3 nanomaterials-10-02119-f003:**
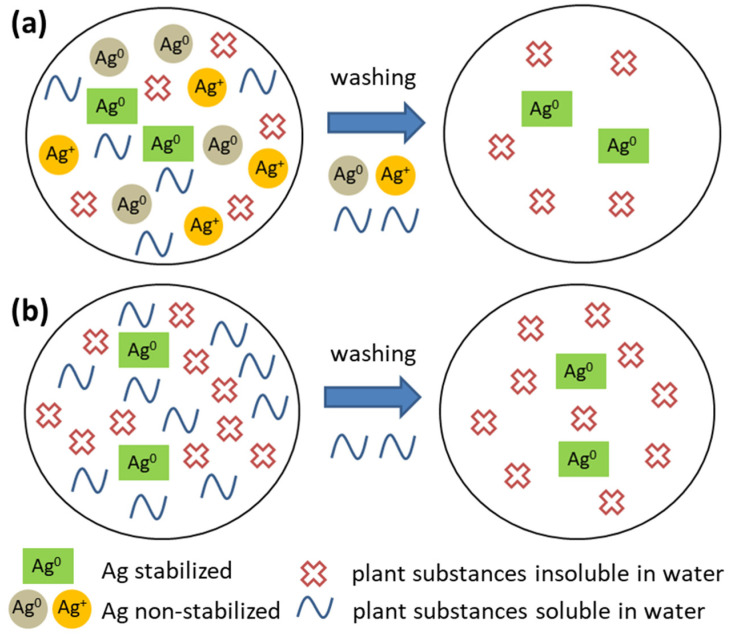
Scheme depicting the washing process for the samples of different AgNO_3_:plant mass ratios: (**a**) 1:1 and (**b**) 1:10–1:100.

**Figure 4 nanomaterials-10-02119-f004:**
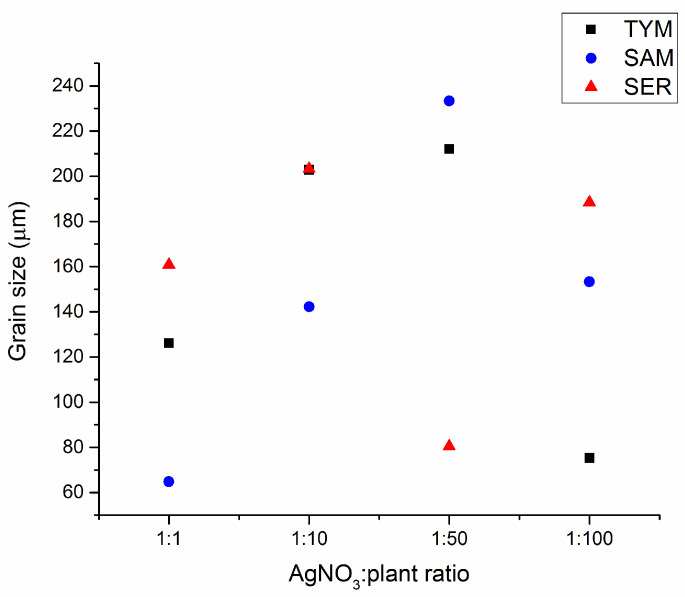
Grain size analysis of Ag:plant washed samples.

**Figure 5 nanomaterials-10-02119-f005:**
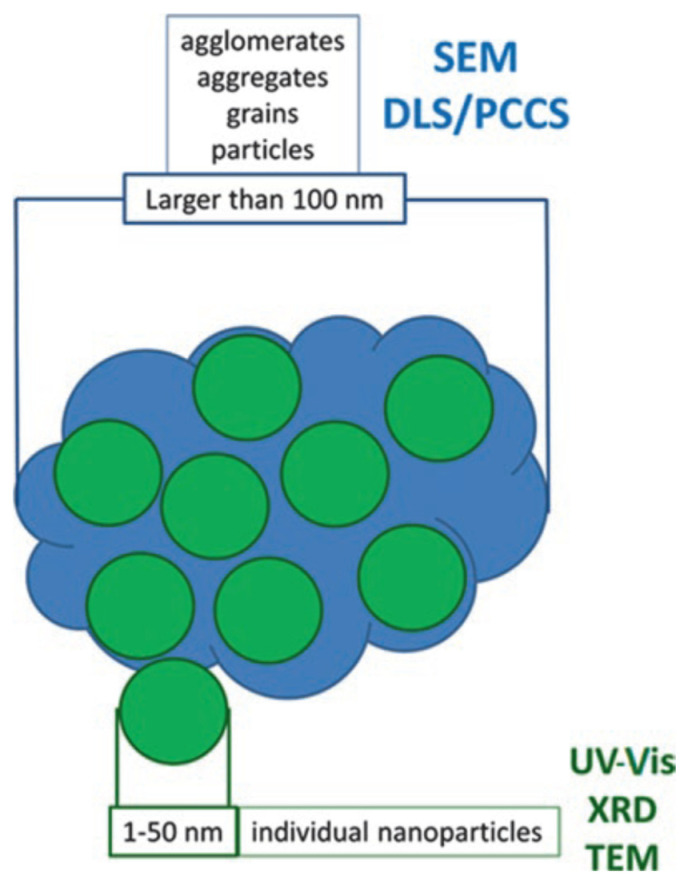
Schematic illustration showing which techniques report the size of individual nanoparticles of few nanometers and that of agglomerates/grains of size larger than 100 nm. Reprinted by permission from Springer Nature Customer Service Centre GmbH: Springer Nature [31], COPYRIGHT (2020).

**Figure 6 nanomaterials-10-02119-f006:**
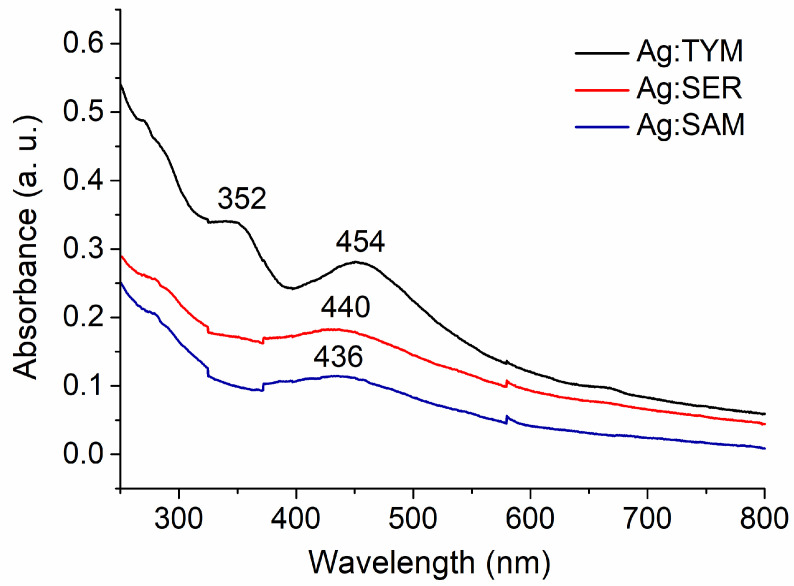
UV-Vis spectra of Ag:plant 1:1 W samples.

**Figure 7 nanomaterials-10-02119-f007:**
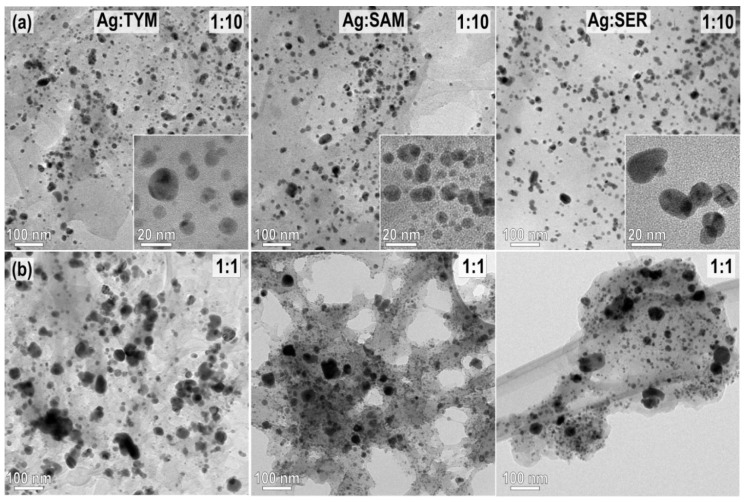
TEM images of the Ag:plant W samples (**a**) 1:10 and (**b**) 1:1. Insets in the (**a**) are magnifications showing the presence of AgNPs with sizes below 10 nm.

**Figure 8 nanomaterials-10-02119-f008:**
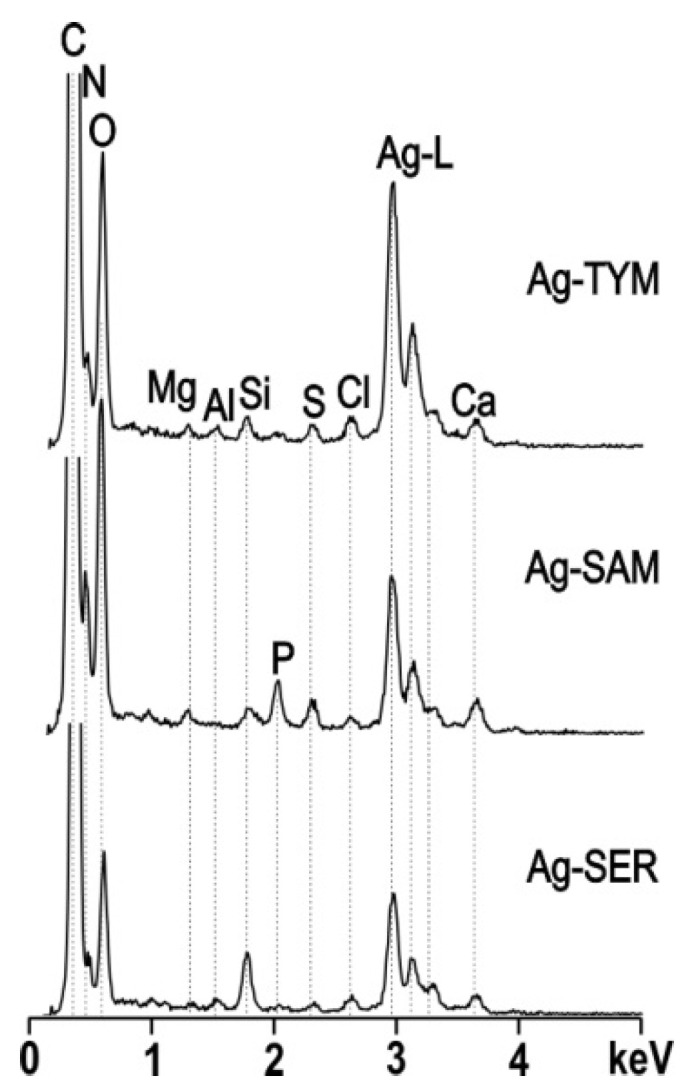
TEM/EDS spectra recorded for the samples with AgNO_3_:plant ratio 1:10. The analyses included both organic matrix and AgNPs.

**Figure 9 nanomaterials-10-02119-f009:**
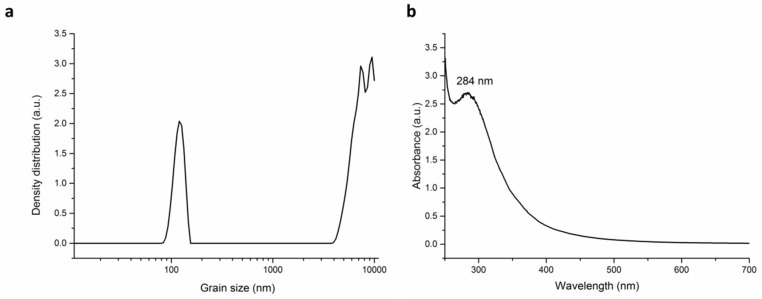
Characterization of the filtrate of Ag:SER 1:1 sample: (**a**) grain size distribution, (**b**) UV-Vis spectrum.

**Figure 10 nanomaterials-10-02119-f010:**
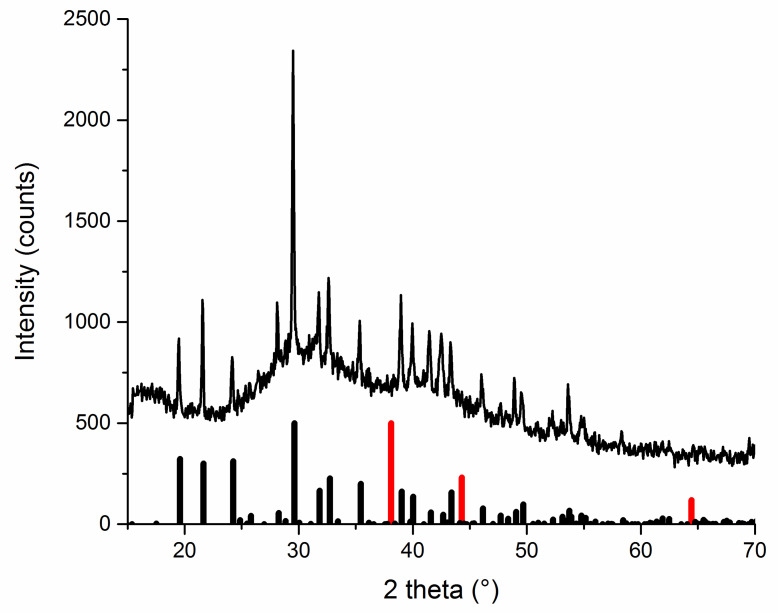
XRD pattern of the dried filtrate after washing of Ag:SER 1:1 sample (bars at the bottom refer to the records in ICDD PDF2 database: AgNO_3_ (black, 074-2076), Ag (red, 65-2871).

**Figure 11 nanomaterials-10-02119-f011:**
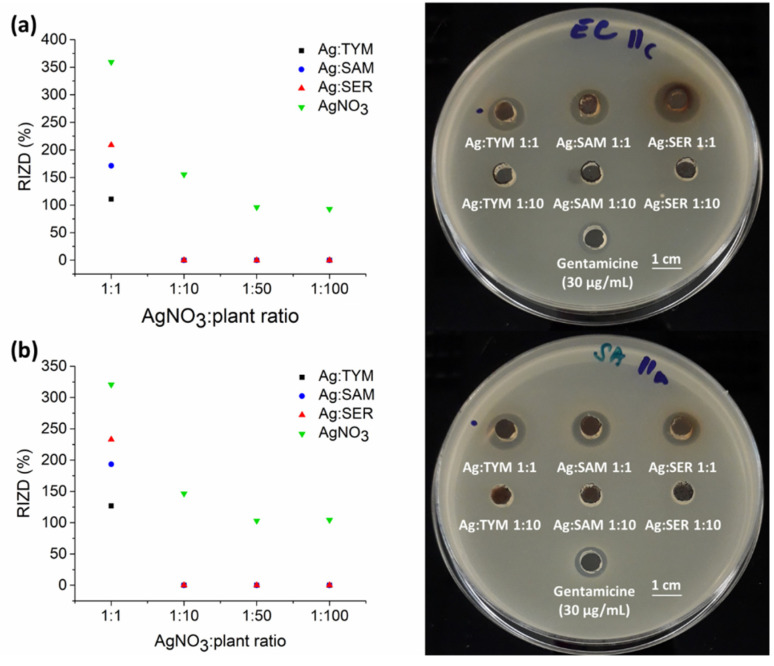
Left: Antibacterial activity of AgNPs-plant samples against: (**a**) *E. coli* and (**b**) *S. aureus*. Right: Representative photographs of Petri dishes after 24 h incubation. RIZD stands for relative inhibition zone diameter.

**Table 1 nanomaterials-10-02119-t001:** Phase analysis of washed Ag:plant 1:1 and 1:10 samples (phase content is in %).

Sample	1:1		1:10		
	Ag	WC	Ag	WC	AgCl
**Ag:TYM**	99.60 ± 0.27	0.40 ± 0.27	78.0 ± 2.3	18.6 ± 2.3	3.4 ± 0.7
**Ag:SAM**	100	-	37.7 ± 6.7	62.3 ± 6.7	-
**Ag:SER**	100	-	93.6 ± 1.6	-	6.4 ± 1.6

**Table 2 nanomaterials-10-02119-t002:** Crystallite size (nm) of washed Ag:plant 1:1 and 1:10 samples.

Sample	1:1	1:10
**Ag:TYM**	22 ± 4	19 ± 2
**Ag:SAM**	18 ± 0.3	13 ± 0.2
**Ag:SER**	27 ± 3	19 ± 4

**Table 3 nanomaterials-10-02119-t003:** Ag until cell volume (Å^3^) for washed Ag:plant 1:1 and 1:10 samples.

Sample	1:1	1:10
**Ag:TYM**	68.101 ± 0.043	68.304 ± 0.074
**Ag:SAM**	68.075 ± 0.059	68.329 ± 0.071
**Ag:SER**	67.982 ± 0.040	68.265 ± 0.078

**Table 4 nanomaterials-10-02119-t004:** Ag content in the solid samples obtained after milling and also after subsequent washing determined using atomic absorption spectrometry.

Sample	Ag Content in the Milled Sample (M) (%)	Ag Content in the Milled Sample after Washing (W) (%)	W/M Ratio
**Ag:TYM 1:1**	31.20	18.34	0.59
**Ag:SAM 1:1**	32.50	23.31	0.72
**Ag:SER 1:1**	32.10	26.06	0.81
**Ag:TYM 1:10**	5.64	10.95	1.94
**Ag:SAM 1:10**	5.40	11.33	2.10
**Ag:SER 1:10**	4.95	12.28	2.48
**Ag:TYM 1:50**	0.60	2.19	3.65
**Ag:SAM 1:50**	0.75	2.28	3.04
**Ag:SER 1:50**	0.22	1.83	8.32
**Ag:TYM 1:100**	0.13	0.80	6.15
**Ag:SAM 1:100**	0.16	0.97	6.06
**Ag:SER 1:100**	0.15	0.63	4.20

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
