# Peer review of "Sustainable One-Step Solid-State Synthesis of Antibacterially Active Silver Nanoparticles Using Mechanochemistry"

_nanomaterials, 2020, doi:10.3390/nano10112119_

Round 1

Reviewer 1 Report

Review

Sustainable one-step solid-state synthesis of antibacterially active silver nanoparticles using mechanochemistry

The authors describe a bio-mechanochemical approach for green Ag-NPs synthesis using different ratios of AgNO3 and three dry plants (Thymus serpyllum L. (SER), Sambucus nigra L.  (SAM) and Thymus vulgaris L. (TYM)). The field of Ag-NPs is already broadly investigated, however, the mechanochemical synthetic approach is seldom described. The synthesized nanoparticles are characterized by a variety of spectroscopic and microscopic techniques. I recommend publishing this contribution with corrections.

Below my questions and comments to improve the manuscript:

Mechanochemical synthesis

R 67-69: The authors use a washing step with water to remove unreacted Ag+ as well as non-stabilized Ag0. Such a step is indeed needed otherwise the antibacterial activity would be significantly influenced by the remaining AgNO3 (which shows higher activity compared to the Ag-NPs). In this step, however, all water-soluble ligands as well as functionalized NPs with such ligands could/would also be removed. Did the authors check using UV-vis spectroscopy (it is fast and cheap, or TEM, XRD, etc.) if some Ag-NPs have actually been removed (also the size and distribution)? If so, additional information is needed as water soluble NPs might have an influence on the antibacterial activity. To separate the soluble NPs from AgNO3 centrifugation could be used.

Results and discussion – Particle size and UV-Vis

I do not see what relevant information brings determining the particle size for the purposes of the paper - these are actually agglomerates in mm range with a lot of organic matrix. How does this information relate with the TEM images or with the antibacterial activity? How would the authors correlate the particle size of the Ag-NPs (in case of 1:1 Ag:TYM 120 mm and Ag:SER 160 mm) and the maximum of the plasmon resonance (454 nm for the former and 440 nm for the latter case). Same question is valid for the size of the crystallites Ag:TYM and Ag:SER. The authors should comment on this discrepancy.

TEM analysis

R 310-311: The authors try to relate the mobility of the NPs with the bimodal size distribution. I do not think that mobility plays a role in a two-hour ball-mill synthesis at which conditions intensive mixing and heating take place. It is more a matter of concentration and type of the ligands in the plant. Also, stating that “a fraction of NPs remain attached to the organic matrix…” is not precise. Actually, the synthesized Ag-NPs are stabilized by the organic matrix (being a matter of concentration, of course).

Antibacterial activity

R 340-344 The results show highest activity of the Ag:SER 1:1 sample. How does this finding correlate with the presented on Fig 4 particle size analysis? It is in my opinion counterintuitive the largest NPs/agglomerates with the smallest surface area to display highest activity.

The activity is probably related with the exposed NP surface (non-functionalized) after washing and most probably with residual AgNO3.

Author Response

Reviewer 1:

Sustainable one-step solid-state synthesis of antibacterially active silver nanoparticles using mechanochemistry

The authors describe a bio-mechanochemical approach for green Ag-NPs synthesis using different ratios of AgNO3 and three dry plants (Thymus serpyllum L. (SER), Sambucus nigra L.  (SAM) and Thymus vulgaris L. (TYM)). The field of Ag-NPs is already broadly investigated, however, the mechanochemical synthetic approach is seldom described. The synthesized nanoparticles are characterized by a variety of spectroscopic and microscopic techniques. I recommend publishing this contribution with corrections.

Below my questions and comments to improve the manuscript:

Mechanochemical synthesis

R 67-69: The authors use a washing step with water to remove unreacted Ag+ as well as non-stabilized Ag0. Such a step is indeed needed otherwise the antibacterial activity would be significantly influenced by the remaining AgNO3 (which shows higher activity compared to the Ag-NPs). In this step, however, all water-soluble ligands as well as functionalized NPs with such ligands could/would also be removed. Did the authors check using UV-vis spectroscopy (it is fast and cheap, or TEM, XRD, etc.) if some Ag-NPs have actually been removed (also the size and distribution)? If so, additional information is needed as water soluble NPs might have an influence on the antibacterial activity. To separate the soluble NPs from AgNO3 centrifugation could be used.

The reviewer is right in this point, so we have repeated the milling experiment and the washing process with the selected sample (Ag:SER 1:1). The obtained filtrate was subjected to photon cross-correlation, UV-VIS and XRD measurements. The obtained results have been included into the manuscript (see Figures 9 and10). It turned out that there is not a significant fraction of AgNPs present in the filtrate, thus the stabilization process seems to be efficient.

 Results and discussion – Particle size and UV-Vis

I do not see what relevant information brings determining the particle size for the purposes of the paper - these are actually agglomerates in mm range with a lot of organic matrix. How does this information relate with the TEM images or with the antibacterial activity? How would the authors correlate the particle size of the Ag-NPs (in case of 1:1 Ag:TYM 120 mm and Ag:SER 160 mm) and the maximum of the plasmon resonance (454 nm for the former and 440 nm for the latter case). Same question is valid for the size of the crystallites Ag:TYM and Ag:SER. The authors should comment on this discrepancy.

The answer for this question can be nicely explained by the figure below which was published in a book chapter of ours this year. In mechanochemical synthesis, we usually obtain nanocrystalline materials, however the individual nanoparticles are agglomerated to micron-sized grains. Whereas TEM, UV-VIS and XRD report the results of crystallite size of individual nanoparticles, SEM, PCCS or DLS methods report the size of grains/clusters/agglomerates (please see the figure below). These two approaches are often incorrectly interchanged in the papers. The reviewer is mistaken in the units of particle size, as they are in micrometer, not in millimeter range. From our point of view, the observed particle size is not related to the TEM images or the antibacterial activity. For the latter, the crystallite size as determined by TEM, XRD or UV-Vis is decisive. We have included the figure below and explained the mentioned facts in the manuscript. We have also introduced the term “grain size” to clearly distinguish the two phenomena.

TEM analysis

R 310-311: The authors try to relate the mobility of the NPs with the bimodal size distribution. I do not think that mobility plays a role in a two-hour ball-mill synthesis at which conditions intensive mixing and heating take place. It is more a matter of concentration and type of the ligands in the plant. Also, stating that “a fraction of NPs remain attached to the organic matrix…” is not precise. Actually, the synthesized Ag-NPs are stabilized by the organic matrix (being a matter of concentration, of course).

We completely agree with the reviewer, so his interpretation has been used in the manuscript and the corresponding text in the TEM part has been modified accordingly.

Antibacterial activity

R 340-344 The results show highest activity of the Ag:SER 1:1 sample. How does this finding correlate with the presented on Fig 4 particle size analysis? It is in my opinion counterintuitive the largest NPs/agglomerates with the smallest surface area to display highest activity.

We are of the opinion that the particle size is not directly connected with the antibacterial effect, as it is mainly governed by the crystallite size (detected by TEM, or XRD, as described earlier). The particle size of the obtained clusters is in hundreds of micrometers, whereas the size of bacteria is around 1 micrometer. The individual nanoparticles with the size of tens of nanometers can penetrate into the cell wall. Moreover, if we imagine how the antibacterial testing is done, it is clear that the coarse agglomerates settle down at the bottom of the well punched in the agar immediately after the injection and only the fraction of small nanoparticles, which is dispersed in the distilled water is then penetrating throughout the agar and killing the bacteria to yield the transparent zone.

The activity is probably related with the exposed NP surface (non-functionalized) after washing and most probably with residual AgNO3.

There is no residual AgNO3 in the final products, as it was washed out during the washing step. As it was proven by the characterization results of the dried filtrate, it contains no AgNPs. Thus, the final activity can be ascribed to the AgNPs stabilized (functionalized) within the organic matrix.

Reviewer 2 Report

the paper is interesting but it needs some revision before publishing.

there are several papers stating the better performances of AgNP towards gram negative bacteria thant towards gram positive strains, and some of them should be cited in addition to ref 14. the fact that gram positive bacteria as S.Aureus are more robust towards AgNP action is widely recognized, and authors should cite some more papers about this fact. see for example  RSC advances  (2016) 6 (74), 70414-70423, European Journal of Inorganic Chemistry 2018 (45), 4846-4855.

A temptative explanation of the reason why only the 1:1 samples are effective as antibacterial should be given

paragraph "3.3. Particle size analysis" is somewhat confusing, as one is induced to think that authors are talkinf about Ag nanoparticles, while, I suppose, this paragraph relates about partciles of plant matrix embedding AgNP

the value of the average size for the NP popultaions found, or even better, some histograms representing the polydispersity, should be given in the TEM paragraph.

authors should also try to compare the values of the size and Uv-vis peaks data with those of ported authors (refs 15, 16 and 18). Probably, the differences in absorption maxima of AgNP when compared to values in the given references are due to differences in mean size of AgNP.

in general, it is somewhat suprising that smaller AgNP (i.e. AgNP coming from higher plant/Ag ratios) have lower antibacterial activity, and authors should comment a bit about this.

Author Response

Reviewer 2:

the paper is interesting but it needs some revision before publishing.

there are several papers stating the better performances of AgNP towards gram negative bacteria thant towards gram positive strains, and some of them should be cited in addition to ref 14. the fact that gram positive bacteria as S.Aureus are more robust towards AgNP action is widely recognized, and authors should cite some more papers about this fact. see for example  RSC advances  (2016) 6 (74), 70414-70423, European Journal of Inorganic Chemistry 2018 (45), 4846-4855.

The reviewer is right that there are many reports showing more pronounced antibacterial effect of AgNPs against gram-negative than against gram-positive bacteria. We have included one reference from the ones suggested by the reviewer and added three more to support this fact.

A temptative explanation of the reason why only the 1:1 samples are effective as antibacterial should be given

The amount of formed AgNPs decreased with the amount of introduced AgNO3. Thus, in the samples where the amount of plant material is 10-, 50- and 100-fold higher (in the samples Ag:plant 1:10, 1:50 and 1:100), the content of AgNPs is too low to show antibacterial properties (for reference, please check the content of silver in the samples detected by AAS- Table 4). This fact has been stressed out in the paper.

paragraph "3.3. Particle size analysis" is somewhat confusing, as one is induced to think that authors are talkinf about Ag nanoparticles, while, I suppose, this paragraph relates about partciles of plant matrix embedding AgNP

The same issue has been raised by the other reviewer.  The confusion can be easily clarified by the figure below which was published in a book chapter of ours this year. In mechanochemical synthesis, we usually obtain nanocrystalline materials, however the individual nanoparticles are agglomerated to micron-sized grains. Whereas TEM, UV-VIS and XRD report the results of crystallite size of individual nanoparticles, SEM, PCCS or DLS methods report the size of grains/clusters/agglomerates (please see the figure below). These two approaches are often incorrectly interchanged in the papers. The reviewer is right that the large grains are that of plant matrix in which AgNPs are embedded. We have included the figure below and explained the mentioned facts in the manuscript. We have also corrected the terminology in the paper and we are using “grain size” when referring to the size of the large grains/agglomerates (blue in the figure below).

the value of the average size for the NP popultaions found, or even better, some histograms representing the polydispersity, should be given in the TEM paragraph.

Based on this reviewer’s comment, a large amount of individual nanoparticles has been measured and the values of average crystallite size for the nanoparticles are now mentioned in the manuscript in the following sentence: „The average crystallite size is always larger for Ag:plant 1:1 samples than for 1:10 samples and no difference has been observed when using different plants. Namely, the crystallite sizes range from < 5 nm to around 40 nm in samples with 1:10 ratio, whereas in the samples with 1:1 AgNO3:plant ratios, the largest AgNPs reach diameters around 80 nm.”

authors should also try to compare the values of the size and Uv-vis peaks data with those of ported authors (refs 15, 16 and 18). Probably, the differences in absorption maxima of AgNP when compared to values in the given references are due to differences in mean size of AgNP.

We have added the following correlation of the positions of SPR peaks detected using UV-VIS spectroscopy with the crystallite size estimated from X-ray diffraction (Table 2) into the paper:

“ The phenomenon that smaller nanoparticles should possess the SPR band at the lower wavelength has been at least partially confirmed for Ag:SAM 1:1 sample, which contained the smallest NPs (18± 0.3 nm) and the SPR band is located at 436  nm (lowest wavelength of the studied samples). However, this correlation did not work for the other two Ag:plant 1:1 systems, as the largest particles evidenced in the case of Ag:SER 1:1 sample (27 ± 3 nm) exhibited the SPR band located at 440 nm and in the case of Ag:TYM 1:1, where the crystallite size was 22 nm (in between the ones of Ag:SER 1:1 and Ag:SAM 1:1), the maximum SPR value from all three studies samples located at the highest wavelength (454 nm) has been observed.”

Moreover, also the discussion with relevant literature on this fact has been provided: “The assumption about increasing the wavelength of SPR maximum in the UV-VIS spectra was also not confirmed when comparing the results from relevant literature, as for AgNPs prepared using TYM, SAM and SER, the SPR peak position was 440, 407 and 467 nm and the crystallite size calculated from TEM was 30, 25 and 25.2 nm, respectively. However, the validity of such a comparison of the results from different studies achieved at different experimental setups is questionable.”

in general, it is somewhat suprising that smaller AgNP (i.e. AgNP coming from higher plant/Ag ratios) have lower antibacterial activity, and authors should comment a bit about this.

Quite similar question of this reviewer has been answered earlier. The amount of formed AgNPs decreases with the amount of introduced AgNO3. Thus, in the samples where the amount of plant material is 10-fold higher (in the samples Ag:plant 1:10), the content of AgNPs (although they are smaller than in the case of Ag:plant 1:1 samples) is too low to show antibacterial properties. It might seem surprising that when comparing the antibacterial activity of three Ag:plant 1:1 samples, the one with the largest NPs (Ag:SER) is the best agent, but again, the amount of Ag in this sample is the highest here (please consult Table 4) , so it seems that not the size, but the content of Ag is decisive in our case. We have included also this explanation into the manuscript.

Reviewer 3 Report

I found this paper very interesting and, in some ways, also new.
Since enough reports have already been acquired by the Editor to be able to judge whether or not to publish the paper, I limited myself to a first reading, without going into too much detail of revision.
My first impression is positive and therefore I would see this paper published with pleasure.
If there should be a second round in the review process, I will be happy to give my suggestions.

Author Response

We thank the reviewer for devoting his/her time to reading our manuscript and for the positive evaluation of our work. We will be glad to receive the suggestions from the reviewer if the editorial office decides that another review report is necessary.